# Early Postnatal Exposure to Midazolam Causes Lasting Histological and Neurobehavioral Deficits via Activation of the mTOR Pathway

**DOI:** 10.3390/ijms25126743

**Published:** 2024-06-19

**Authors:** Jing Xu, Jieqiong Wen, Reilley Paige Mathena, Shreya Singh, Sri Harsha Boppana, Olivia Insun Yoon, Jun Choi, Qun Li, Pengbo Zhang, Cyrus David Mintz

**Affiliations:** 1Department of Anesthesiology and Critical Care Medicine, The Johns Hopkins University School of Medicine, Baltimore, MD 21209, USA; jxu72@jhmi.edu (J.X.); wenjieqiong1992@163.com (J.W.); r.paigemathena@gmail.com (R.P.M.); shreya.singh4876@gmail.com (S.S.); sboppan2@jh.edu (S.H.B.); xuj.1988@gmail.com (J.C.); qli21@jhmi.edu (Q.L.); 2Department of Anesthesiology, The First Affiliated Hospital of Xi’an Jiaotong University School of Medicine, Xi’an 710061, China; 3Department of Anesthesiology, The Second Affiliated Hospital of Xi’an Jiaotong University School of Medicine, Xi’an 710000, China; zhpbo@mail.xjtu.edu.cn

**Keywords:** midazolam, long-term effects, sedation, mTOR pathway, dendrite morphology

## Abstract

Exposure to general anesthetics can adversely affect brain development, but there is little study of sedative agents used in intensive care that act via similar pharmacologic mechanisms. Using quantitative immunohistochemistry and neurobehavioral testing and an established protocol for murine sedation, we tested the hypothesis that lengthy, repetitive exposure to midazolam, a commonly used sedative in pediatric intensive care, interferes with neuronal development and subsequent cognitive function via actions on the mechanistic target of rapamycin (mTOR) pathway. We found that mice in the midazolam sedation group exhibited a chronic, significant increase in the expression of mTOR activity pathway markers in comparison to controls. Furthermore, both neurobehavioral outcomes, deficits in Y-maze and fear-conditioning performance, and neuropathologic effects of midazolam sedation exposure, including disrupted dendritic arborization and synaptogenesis, were ameliorated via treatment with rapamycin, a pharmacologic mTOR pathway inhibitor. We conclude that prolonged, repetitive exposure to midazolam sedation interferes with the development of neural circuitry via a pathologic increase in mTOR pathway signaling during brain development that has lasting consequences for both brain structure and function.

## 1. Introduction

The U.S. Food and Drug Administration (FDA) has published a Drug Safety Communication that warns that repeated or lengthy exposure to anesthetic and sedative medications in young children may have harmful effects on brain development [1,2]. These concerns stem from large epidemiologic studies showing a correlation between worsened cognitive outcomes for patients who underwent surgery and anesthesia at a young age, rodent model investigations showing that general anesthetics can disrupt many aspects of brain development, and primate studies demonstrating a variety of cognitive and behavioral deficits resulting from early anesthetic exposure in development [3,4,5,6,7,8,9,10,11]. Clinical trials have provided reassurance that short exposures to general anesthesia, typically an hour or less, in healthy children do not cause a measurable reduction in intelligence, although it appears increasingly likely that there are deleterious effects on other domains of neurobehavioral function [12]. However, in the intensive care unit (ICU) setting, children are typically sedated for many hours, and in some cases for days or even weeks at a time [13,14]. Neurologic impairment unrelated to the underlying disease is an unfortunately common outcome associated with critical illness in children [15], and concerns have been raised in the pediatric critical care literature that exposure to sedative medications may be an iatrogenic cause of worsened neurocognitive and neurobehavioral outcomes in ICU patients [16,17,18].

While concerns about sedation toxicity, particularly in relation to midazolam, have been raised in the pediatric critical care literature, there has been little investigation specifically directed towards sedative effects on brain development to date. Midazolam, a benzodiazepine that is commonly delivered either orally, as a single-dose medication or as a continuous infusion, is among the most commonly used sedative agents in pediatric critical care [13,14]. Midazolam and lorazepam were both selected for label changes indicating the FDA communication related to anesthetic and sedative medications [1,2]. At the time of the FDA warning, there were three main lines of evidence raising concerns regarding the use of midazolam in infants and children: 1. the shared general mechanism of action on γ-aminobutyric acid (GABA) receptors between midazolam and better-studied anesthetics such as the potent volatile anesthetics and propofol [19]; 2. the inclusion of midazolam as one agent in a cocktail of commonly used drugs has been studied together in some animal model studies of developmental anesthetic neurotoxicity, e.g., [20]; and 3. the common practice of premedication with midazolam in pediatric anesthesia [21], which leads to the assumption that it is represented in epidemiologic studies of pediatric anesthetic neurotoxicity. Midazolam exposure may lead to acute behavioral changes, such as alterations in anxiety-like behavior, social interactions, or exploratory behavior, in young mice. These effects have been assessed through behavioral assays tailored to measure specific endpoints [22,23]. In our previous research, we showed that late postnatal mice exposed to midazolam in a sedative paradigm exhibit chronic, lasting deficits in performance on behavioral tests of learning and memory, and both in vivo and in vitro analyses showed that lengthy exposure to midazolam resulted in disruptions of neurogenesis and synaptogenesis [24]. While no clinical investigations have been conducted to test the relatively novel hypothesis that early-life midazolam exposure in an intensive care sedative paradigm can cause harm to brain health, the two other extant animal model studies broadly support this conclusion [25,26]. Furthermore, the molecular and cellular mechanisms by which midazolam might disrupt brain development remain unclear.

The mammalian target of rapamycin (mTOR) proteins are kinases that form complexes at the nexus of an information processing system that integrates a broad array of intracellular and extracellular signals to regulate cellular function. The mTOR pathway plays a critical role in brain development [27,28], and disruptions in mTOR signaling have been associated with neurodevelopmental disorders [29,30,31]. Our previous work has shown that transient exposure to the anesthetic isoflurane, which, like midazolam, is a potent GABA agonist, causes a chronic upregulation of mTOR pathway activity that interferes with brain development [32,33,34]. Here, we test the hypothesis that early developmental exposure to midazolam in a sedative paradigm causes lasting neurobehavioral deficits and disruption of dendritic and synaptic development via a chronic effect on mTOR signaling.

## 2. Results

### 2.1. mTOR Pathway Activity

The effects of an ICU-relevant paradigm of midazolam administered on postnatal days 18 through 22 (P18–22) and titrated to deep sedation on a validated rating scale [24] on mTOR pathway activity in the dentate gyrus were tested (Figure 1A). On P63, cell bodies with immunoreactivity for phospho-mTOR and phospho-S6, both of which are mTOR activity markers, in the dentate gyrus (DG) were counted, and the DG area was measured to allow for density measurements (Figure 1B,C). The results showed that both the phospho-mTOR and phospho-S6 positive cell density in the dentate gyrus of the midazolam-exposed groups were increased significantly compared to the control group (phospho-mTOR: 1141 ± 58.53% in the midazolam group compared to 665.7 ± 63.75% in the control group; phospho-S6: 1296 ± 98.42% in the midazolam group compared to 514.9 ± 78.51% in the control group, *p* < 0.0001) (Figure 1D,E). To confirm that the activation of these mTOR downstream markers was induced by midazolam, either vehicle control or 20 mg/kg rapamycin, a pharmacologic inhibitor of mTOR, was given through intraperitoneal injection every other day from P23 to P31. The data showed that the rapamycin treatment significantly reduced phospho-mTOR and phospho-S6 immunoreactivity compared to midazolam groups and that levels were comparable to the untreated controls (phospho-mTOR: 630.8 ± 76.8% in the midazolam and rapamycin group compared to the control group, no significant difference; phospho-S6: 458.9 ± 39.15% in 20 mg/kg midazolam and rapamycin group compared to the control group, no significant difference) (Figure 1D,E). Taken together, these data provide strong evidence that midazolam sedation during early postnatal life results in a chronic upregulation of mTOR pathway signaling.

### 2.2. Neurobehavioral Outcomes

Behavioral testing was performed at P63, which coincides with histological measurements of mTOR pathway activity already shown and measurements of dendrite and synapse development shown below (Figure 2A). Rapamycin treatment after midazolam exposure was conducted on P23–31, as this was shown to be effective in preventing lasting mTOR activation.

The Y-maze test was used to evaluate sedation toxicity effects on spatial learning. It is highly sensitive to the small alternations in DG neurons [35]. The control animals spent more time exploring the novel arm (59.01 ± 10.34%) than the original arm (40.91 ± 10.34%, *p* < 0.01), as expected in normal mice, which were unimpaired in their ability to remember which arm they had visited previously. The animals in the midazolam sedation group did not exhibit a preference for exploration of the newly opened arm compared to the control group. With rapamycin treatment, this phenomenon was reversed, and the midazolam-sedated animals behaved similarly to the control animals, with a significantly longer exploration time inside the new arm compared to the old one (Figure 2B,C).

To provide supporting data on the effects of midazolam toxicity on learning and memory functions, contextual and cued fear-conditioning tests were conducted (Figure 2D,E). This paradigm differs from the Y-maze test in several ways, including a lack of dependence on locomotor or visual function. In the contextual test, when compared to the controls, midazolam-sedated mice showed normal freezing time percentages after the rapamycin treatment (Figure 2F). In the cued test, midazolam-sedated mice showed a decreased freezing time comparable to the control animals, and this decreased performance was mitigated by treatment with rapamycin (Figure 2G). Taken together, these findings indicate that lasting neurobehavioral abnormalities caused by early postnatal midazolam sedation are dependent on a midazolam-induced increase in mTOR pathway activity.

### 2.3. Dendrite Development

To investigate the effects of sedation drugs on dendrite development in vivo, we employed stereotaxic injection to deliver a retrovirus expressing green florescent protein (GFP) to label the newly generated neurons in the dentate gyrus [36]. The injections were carried out at P15. A rapamycin treatment group, conducted as above, was included to test for the effects of mTOR activity normalization. Dendritic structure was analyzed on P63 by tracing the dendritic arbors of GFP-labeled neurons in the DG (Figure 3A).

As shown in Figure 3B,C, the GFP-labeled neurons have readily visible dendritic arbors, which were easily traced and analyzed. Measurements of total dendrite arbor length showed that in comparison to the neurons of unexposed control littermates, the labeled neurons in the midazolam-exposed animals exhibited a significant mean 37.18% decrease (325.23 ± 101.47 μm compared to 517.68 ± 114.36 μm in the control group, *p* < 0.0001). In the rapamycin treatment group (475.10 ± 109.24 μm), there was no significant change in dendritic arbor length compared to the control group (Figure 3E). We also examined whether the branch numbers of the neurons were altered by midazolam exposure. The analysis showed a significant 30.95% decrease in the mean total branch numbers in the midazolam-exposed group (3.32 ± 1.69 compared to 4.80 ± 2.20 μm in the control group, *p* < 0.001). With rapamycin treatment, the branch numbers (4.47 ± 2.36) showed no significant difference compared to the control group (Figure 3F,G). To evaluate dendritic arbor complexity, we performed Sholl analysis. We found that the number of crossings was significantly decreased compared to controls in the 150 and 160 μm rings and significantly increased in the 210 to 230 μm rings (*p* < 0.05) (Figure 3H,I). When mTOR activation induced by midazolam was inhibited with rapamycin, the effects of midazolam were replaced by a distinct change in the pattern of complexity, indicating that rapamycin had an independent effect (Figure 3H,I). These findings of reductions in length, branching, and concentric crossings represent a substantial disruption of dendrite development based on comparisons to other studies of dendrite arborization [37,38,39]. They were largely reversed with rapamycin treatment, which indicates a dependence on mTOR pathway upregulation.

### 2.4. Synapse Development

Next, we explored the effects of midazolam sedation and rapamycin inhibiting mTOR signaling upregulation on synapse formation (Figure 4A). We used the same retrovirally labeled populations of neurons shown in Figure 3 to examine the effects on the density of dendritic spines, which represent excitatory synapses, and the subgroup of spines that exhibit mature mushroom morphology, which is correlated with memory consolidation functions. While there was no significant difference in the total density of the spines in the midazolam-exposed group (2.08 ± 0.56/μm) or the control group (2.10 ± 0.55/μm, *p* = 0.9876), there was a significant decrease in the density of the mature mushroom spines (midazolam: 0.20 ± 0.11/μm vs. control: 0.26 ± 0.11/μm, *p* < 0.05) (Figure 4A–C). This represents a 23.1% decrease in the population of spines, which are believed to be primarily responsible for the storage of long-term memory [40], and one that is comparable to other investigations of mushroom spine loss that have demonstrated physiological significance [41,42,43]. Treatment with rapamycin after midazolam exposure restores mushroom spine density so that it is no longer significantly different from control levels (Figure 4A,C).

The density of inhibitory synapses was quantified using immunolabeling for the inhibitory synapse marker gephyrin, which allowed for density measurements of gephyrin-positive puncta that represent inhibitory synapses (Figure 4D). With midazolam sedation, there was a significant decrease in gephyrin puncta density compared to the control group (midazolam: 0.20 ± 0.11/μm^2^ vs. control: 0.33 ± 0.13/μm^2^, *p* < 0.01) (Figure 4E). This corresponds to a 40.56% decrease in inhibitory synapse density. Rapamycin treatment prevented the effect of midazolam on inhibitory synapses (midazolam plus rapamycin: 0.39 ± 0.15/μm^2^ vs. control: 0.33 ± 0.13/μm^2^, no significant difference). The role of inhibitory synapses in learning and memory has been well established, but newer literature suggests that inhibitory synapses may also be important for learning functions as well [44], suggesting that effects on both excitatory and inhibitory synapse populations may contribute to the harmful effects of midazolam sedation on brain development.

## 3. Discussion

Here, we sought to test the overall hypothesis that prolonged midazolam exposure designed to model sedation in a pediatric ICU setting during early postnatal life results in neurotoxicity as a result of changes in signaling in the mTOR pathway. We found that midazolam exposure caused a lasting pathologic upregulation of mTOR activity. Furthermore, neurobehavioral deficits as a result of neuropathology, including aberrant dendrite development and synapse loss in the hippocampal dentate gyrus, were reversible via treatment with a pharmacologic mTOR pathway inhibitor, rapamycin (Figure 5).

The U.S. FDA published an advisory that noted that lengthy and/or repeated exposure to general anesthetic agents might have harmful effects on brain development, and this communication called for further research into this area [1,2]. The literature on early developmental anesthetic-induced neurotoxicity incorporates rodent, primate, and human studies, and the mechanisms of injury, exact phenotype, risk factors, and clinical significance remain the subject of ongoing debate and research [5,32,33,45,46,47,48,49,50,51,52]. Many of the general anesthetic agents noted in the FDA advisory share the feature of potent GABA agonism, and both midazolam and propofol, which are frequently used as sedatives rather than general anesthetics, fall into this class of medications. This raises the question of whether the use of these medications may have neurotoxic properties when used as sedatives, and in particular, midazolam finds considerable usage as a sedative in pediatric intensive care settings [53,54]. It is, however, not a foregone conclusion that sedation and general anesthesia will have similar outcomes, given that the drugs used and the dose, timing, pattern, and duration of exposure are considerably different between the intensive care sedation and operating room general anesthesia paradigms. Therefore, most consideration of this topic in relation to patient care has been speculative, but it is notable that the clinical literature on pediatric intensive care contains numerous instances of concerns being cited regarding the neurodevelopmental consequences of early-life sedative exposure [16,17,18,55].

The most direct evidence for early developmental sedation-induced neurotoxicity that addresses both pathologic/mechanistic effects and demonstrates a functional deficit has been attained using rodent models [24,25]. While the rodent model is ideal for investigating mechanisms and exploring possible phenotypes, it does contain the potential bias that studies showing no effect are unlikely to be published. Also, rodent models’ utility is constrained by the considerable differences between human and rodent brain structure and development. Nevertheless, these data deserve strong consideration as the only direct information pertaining to sedative neurotoxicity during development. Our group published a report using a model system identical to the one used in the current study, in which we found that repeated midazolam sedation resulted in learning and memory deficits that were accompanied by alterations in neurogenesis and synaptogenesis [24]. Using a very similar model, Doi et al. further explored the effects of midazolam sedation on ongoing neurogenesis in dentate gyrus granule cell neurons, which revealed that the percentage of actively dividing neural stem cells was reduced by midazolam treatment when assayed 8 weeks later, suggesting that the sedation exposure causes persistent stem cell quiescence [25]. Transposase-accessible chromatin sequencing in dentate gyrus neural stem cells demonstrated a reduced accessibility of stem cell proliferation-related genes that was established after midazolam exposure and persisted to adult age. These changes in neurogenesis were reversible if mice were provided with an exercise wheel, which is a well-characterized stimulus for dentate gyrus neurogenesis. In this investigation, we did not assay neurogenesis as a potential target for sedation neurotoxicity, but our findings are generally consistent in terms of the overall phenotype.

There are relatively few other studies of sedation-induced neurodevelopmental toxicity beyond those referenced above, and direct comparison of these studies to our work is challenged by substantial differences in the sedation model. O’Meara and colleagues used a combination of midazolam and morphine given twice daily for six days starting at P18 in rats. Western blotting of cerebral samples showed an increase in S100 calcium-binding protein B and myelin basic protein levels, but no significant change in synaptophysin, drebrin, or glial fibrillary acid protein expression [26]. Nguyen at al employed a different paradigm with midazolam sedation given daily in escalating doses to mice from ages P3 to P21. They collected synaptosomes from the cerebral cortex and performed mass spectrometry, which revealed differences in expression of a large number of proteins, including those identified by gene ontology analysis as being involved in actin binding, cytochrome C oxidase metabolism, pyridoxal phosphate binding, protein depolymerization, tricarboxylic cycle metabolism, and neuron development [56]. Xu et al. bath-applied midazolam to zebrafish embryos and conducted imaging in the spinal cord, which revealed deficits in oligodendrocyte precursor development that led to decreased myelination. Interestingly, these changes were not prevented by GABA receptor antagonists, but could be mimicked by application of translocator protein [57]. Cabrera et al. found that mice treated at P3 with a single dose of midazolam and caffeine showed an increase in immunohistochemical labeling for activated caspase 3 in multiple brain regions, but that midazolam alone did not show evidence of increased apoptosis [58]. Soyalp et al. administered a single dose of midazolam at P7 in rats and found a decrease in procaspase 3 and caspase 3 levels by Western blotting, suggesting a concurrence with the report above in that midazolam does not seem to cause apoptotic cell death, as has been reported in some early developmental anesthesia exposure models. Interestingly, this group also reported a significant increase in overall oxidative stress index levels on spectrophotometry, which suggests a potential mechanism for a wide array of findings, including our own [59]. In a study of ketamine neurotoxicity in a fetal rat model, Li et al. actually found that midazolam administration reduced expression of markers associated with apoptosis and autophagy in the hippocampus, suggesting a potential neuroprotective effect of midazolam [60]. However, in an in vitro study, Sinner et al. found that E15 rat neurons treated with midazolam exhibited an increase in expression of BAX, Bcl-2 and caspase 3 measured by Western blotting, which is at odds with the findings above, but likely explained by substantial differences in models. In the in vitro system, there were also substantial changes in a variety of synaptic markers, but they did not persist when the midazolam was washed out [61], suggesting a lack of long-term impact, as was found in our experiments. Taken together, the extant literature broadly supports the concept that synaptic development can be disrupted by midazolam sedation in early development, but that cell death is a less likely mechanism of injury. Our findings of reduced mushroom spine morphology and stable total spine numbers indicate the likelihood that an increased proportion of other morphologies results from midazolam exposure, which implies an impairment in excitatory synaptic maturation.

The mTOR pathway is an intriguing potential mechanism of injury in sedation-induced developmental neurotoxicity, as it has been implicated both in normal functions in brain development and disarrayed in a wide range of human neurodevelopmental disease [62]. It is involved in the normal development of dendrites and synapses through its actions, integrating signals from the PI3K–Akt system, which is influenced by both activity and neurotrophic growth factors, such as brain-derived neurotrophic factor (BDNF). In our previous work on anesthetic toxicity in the developing brain, we identified chronic, pathologic upregulation of the mTOR pathway as a cause of neuropathologic changes in synapse formation, dendrite arbor development, and myelination and that these changes and a range of neurobehavioral outcomes that are affected by anesthetic toxicity can be reversed with pharmacologic mTOR inhibition [32,33,63,64,65]. Superficially, these results and those of the current study might be seen to be at odds with a report showing an increase in expression of the mTOR activity marker pS6 in the hippocampus after sevoflurane exposure [66]. However, that study examined only acute effects at four hours after exposure, whereas our work is entirely focused on chronic time points, and the current study examined two different mTOR activity markers at day 63 after exposure, when the animal is reproductively mature and age corresponds to late juvenile or early adult. Our focus on this age corresponds to the observed effects of anesthetic toxicity in epidemiologic studies, which have generally been noted during the teenage years for exposures that occurred in the first four years of life [67,68,69]. Based on our current understanding, we postulate that early-life midazolam exposure causes an upregulation in mTOR signaling, mediated by currently unknown mechanisms. This dysregulation disrupts neurodevelopmental processes, including dendrite arborization and synaptogenesis such that there are substantial neurobehavioral effects (Figure 5). Our work focused on a narrow sample of potential histological and neurobehavioral effects, and further study should be devoted to exploring other potential developmental deficits in brain development that might be caused by similar or perhaps entirely distinct molecular mechanisms.

In conclusion, based on our findings and those in the extant related literature, we propose that early developmental exposure to midazolam causes a lasting, harmful increase in mTOR signaling that disrupts the formation of neural circuitry and results in impairments in neurologic function. In this single-agent study, we considered only midazolam, as it has the highest likelihood of toxicity based on the known literature. Intriguingly, there is increased off-label use of dexmedetomidine for this purpose [13,14], and in contrast to midazolam, dexmedetomidine has been demonstrated to have neuroprotective properties in the setting of anesthesia toxicity in the developing brain [70]. Future study might be directed towards testing whether dexmedetomidine is the ideal choice for pediatric sedation in terms of preserving brain health. Another interesting question related to clinical practice is whether our findings are limited to doses resulting in deep sedation or whether lower levels of midazolam administration might also have harmful consequences. While we do not have a complete dataset to address this question, we did find evidence of increased mTOR activity and impaired outcomes in neurobehavioral testing with a midazolam dose that results in moderate sedation, which is shown in the Appendix A. Further investigation in this area should be directed toward understanding the thresholds in terms of duration and dose that are associated with neurotoxic effects of midazolam sedation during development. Our model system is limited by the short developmental timeline and relatively limited complexity of mouse brain development in comparison to the human equivalent. Also, we considered the effects of sedation alone in the absence of comorbid disease, which would inevitably be present in pediatric patients exposed to lengthy sedation. Even with these limitations taken into account, we believe our findings suggest that further investigation into the potentially harmful effects of sedation on brain development are warranted in both the basic science and clinical arenas.

## 4. Materials and Methods

### 4.1. Ethics

All study protocols involving mice were approved by the Animal Care and Use Committee at Johns Hopkins University (protocol M017M229) and in accordance with the NIH guidelines for the care and use of animals.

### 4.2. Animal Treatment and Handling

#### 4.2.1. Animals

C57BL/6 mice were housed in a temperature-controlled and humidity-controlled room with a 12:12 h light: dark cycle and provided ad libitum access to water and food. Both sexes were equally represented in all experiments. No animals were excluded.

We performed behavioral tests in 6 male and 6 female mice per group. For immunohistochemistry (IHC), we harvested brains using 3 male and 3 female mice per group. For dendrite morphology experiments, we harvested brains using 2 male and 2 female mice per group.

#### 4.2.2. Midazolam Treatment and Physiological Monitoring of Sentinel Animals

The exposures were conducted during daytime hours (07:00 to 19:00) and repeated for 5 days, from P18 to P22. The experimental room was temperature-controlled so that it had the same temperature and humidity compared with the animal room. During the experiment, mice received periodic intraperitoneal injections according to our previously published protocol [24] and were kept on a warming pad maintained at ~38 °C for the duration of the experiment, as well as the recovery time. Exposures were performed starting from P18. Male (weight 7.40 ± 1.55 g) and female (weight 7.35 ± 1.04 g) animals were equally represented. This age was selected because it roughly equates to early childhood in terms of brain development [71] and because mice were robust enough to show minimal physiological and nutritional perturbations in our exposure model. The half-life of midazolam in mice is generally around 1.5 to 2.5 h [71,72,73]. A dose of 20 mg/kg was chosen based on pilot studies to determine what dosing was required to achieve moderate and deep sedation based on the standard rating scale [24,74,75]. The duration of exposure, P18 to P22, which was selected based on pilot experiments shoswing what duration could be used without causing overt signs of nutritional or physiological perturbation (assessed by changes in weight, heart rate, oxygen saturation, and temperature) [32].

Littermates were randomly assigned to 2 groups: group 1 (naïve control): mice were separated from the dams for 12 h, but did not receive injections; and group 2 (midazolam 20 mg/kg): mice separated from the dams were treated with 20 mg/kg midazolam (Hospira Inc., Lake Forest, IL, USA) per hour. The injection was given on a clean bench in a sterile manner. Midazolam was mixed with sterilized saline to adjust it to a uniform volume of 100 μL/10 g b.w. per mouse. The drug was prepared before each injection and kept at the same temperature as the mouse body temperature (38 °C) until administration. After completion of the experiment, mice recovered on the heating pad (Kent Scientific, Torrington, CT, USA) and were returned to their home cages upon regaining righting reflex. Mice were monitored every 15 min, and skin temperature, heart rate, and O_2_ saturation were measured every 30 min during the 12-h midazolam treatment (PhysioSuite, Kent Scientific, CT, USA). Measurements for the control mice were not obtained due to difficulty securing unsedated pups in the PhysioSuite unit, as described earlier [24]. Weights for control and experimental mice were recorded and did not differ significantly.

#### 4.2.3. Measuring Sedation Scores

Sedation levels were measured using the modified grading system described by Kawai et al. [74] and Kirihara et al. [75]. Measurement was based on 4 reflex scales, which are described in detail in our previous study [24]. The total sedation score was graded from 0 to 9 (scoring of control animals maintained at 0). The mice in the 20 mg/kg midazolam groups scored consistently between 5 to 7, which was considered to indicate a deep sedative level in mice [24].

#### 4.2.4. Rapamycin Treatment

P23 mouse littermates were given IP injections of rapamycin (Sigma, St. Louis, MO, USA) prepared from a stock solution (25 mg/mL in 100% ethanol, stored at −20 ℃) diluted to a final concentration of 4% (*v*/*v*) ethanol in the vehicle. Vehicle consisted of 5% Tween 80 (Sigma, St. Louis, MO, USA) and 10% polyethylene glycol 400 (Sigma, St. Louis, MO, USA) as previously described [32,76]. Both rapamycin- and vehicle-treated mice received the same volume for each injection (200 μL). Mice received treatments at 48-h intervals from P23 to P31.

#### 4.2.5. Production and Stereotaxic Injection of Engineered Retroviruses

Engineered self-inactivating murine retroviruses were used to express GFP under the ubiquitin promoter (pSUbGW vector) specifically in proliferating cells and their progeny [32,77]. High titers of engineered retroviruses (1 × 10^9^ unit/mL) were produced by co-transfection of retroviral vectors and VSVG into HEK293gp cells followed by ultracentrifugation of viral supernatant as previously described [77,78]. After induction with a mixture of ketamine 50 mg/mL, high titers of GFP-expressing retroviruses were stereotaxically injected into the P15 mice dentate gyrus through a 32-gauge micro syringe Hamilton, Reno, NV) at 2 sites of the following coordinates relative to the bregma (mm): AP: −2.2, ML: ±2.2, DV: −2.4. The retrovirus-containing solution was injected at a rate of 0.05 μL minute-1 for a total of 0.5 μL per site. After infusion, the micro syringe was left in place for an additional 5 min to ensure full virus diffusion and to minimize backflow. After surgery, mice were monitored for general health every day until full recovery for further experimentation [32].

#### 4.2.6. Behavior Tests

Mice were housed in standard conditions until P60 and then handled by the experimenter for at least 2 min per day for 3 days before the start of the behavioral experiments. Behavior was tested at P63, at which time mice are reproductively mature and brain development roughly correlates with early adulthood [73]. All behavioral tests were performed in the animal behavior core at Johns Hopkins School of Medicine during the light phase of the cycle between 08:00 and 18:00. Experimenters were blind to the condition when behavioral tests were carried out and analyzed. The Y-maze and the fear-conditioning test, two commonly used behavior tests for measuring spatial memory function that are based in the hippocampus, were performed [24,32].

The Y-maze test was performed as previously described [24,32]. In short, in the first phase, a mouse was released from the start arm (contained no visual cue) and allowed to explore only 1 of 2 possible choice arms (each choice arm contained an overt visual cue, red or green) for 15 min (Figure 2B). In the recognition testing phase, which occurred 24 h later, the animal was released from the start arm and had access to both arms for 5 min. The testing phase was video-recorded and the percentage of exploration time in each arm was measured by an observer blind to condition.

The fear-conditioning test was performed according to a commonly used paradigm (Figure 2D) [79]. On the day of the test, the mice were allowed to acclimate to the room for 1 hour before the assay. Each mouse was then placed into the tone–shock chamber (Coulbourn Instruments, Holliston, MA, USA, Figure 2E). The training paradigm involved 2 min of acclimation in the chamber before a 30-s tone, which was followed by a 0.5 mA shock. Then, there was a 2-min consolidation period, followed by a second tone–shock pairing, and this process was repeated three times. The criteria for the exclusion of a mouse from testing was immobility for >30% of the pre-shock habituation/acclimation phase to avoid misinterpreting low habitual mobility as freezing. However, all the animals showed normal mobility during the experiment. Freezing was defined as the complete absence of motion, including motion of the vibrissae, for a minimum of 0.5 s. At 24 h after training, the mice were reintroduced into the same chamber for a period of 5 min to quantify contextual fear conditioning (both the people conducting and recording the experiment were blinded to the treatment condition).

Approximately 1 h following the contextual assay, mice were placed into a novel apparatus that contained a 5% acetic acid scent, where a tone was played for a period of 1 min. In both situations, the outcome measure was the amount of time spent freezing in response to the context or cue. The testing phase was video-recorded, and the percentage of freezing time was measured by FreezeScan software (FreezeScan, 2.18, Clever Sys Inc., Reston, VA, USA). This software assesses freezing by measuring changes in the intensity of each pixel between successive frames of a video file.

### 4.3. Immunohistochemistry

Fluorescent immunocytochemistry for in vivo experiments was performed in accordance with a previous protocol [24]. At P63, the animals were transcardially perfused with 20–30 mL of ice-cold 0.1 M PBS (pH 7.3). Brains were removed and then fixed in 4% paraformaldehyde in PBS at 4 °C overnight, followed by cryoprotection in 30% sucrose in PBS at 4 °C until the tissue sank to the bottom of the vials. The brains containing the dentate gyrus of hippocampus were coronally sectioned (50 μm thickness) using a microtome (Leica SM2010 R sliding microtome, Buffalo Grove, IL, USA). The sections of each brain were collected in rotating order and stored at −20 °C in antifreeze media (30% ethylene glycol and 15% sucrose in PBS) in a 96-well cell culture dish. Five to six sections were picked evenly from the anterior, middle, and posterior part of the hippocampus for future staining. The sections were first washed in PBS (3 × 5 min), followed by blocking for 60 min in 10% normal donkey serum (NGS) and 0.1% Triton X-100 in PBS, after which sections were incubated in primary antibodies at 4 °C for 24 h.

Primary antibodies used in this study were: mouse anti-mTOR (1:50, Invitrogen, Thermo Fisher Scientific, Waltham, MA, USA), rabbit anti-pS6 (1:1000, Fisher Scientific, Hampton, NH, USA), goat anti-GFP (1:1000, Limerick, Rockland, PA, USA), and rabbit anti-gephyrin (1:500, Synaptic Systems, Goettingen, Germany). After 3 × 5-min washes in PBS, sections were incubated with the following secondary antibodies: a combination of Alexa Fluor 488- or Alexa Fluor 594-labeled anti-rat, anti-rabbit, or anti-mouse secondary antibodies (1:250, Jackson ImmunoResearch Inc., West Grove, PA, USA) and 4′,6′-diaminodino-2-phenylindole (DAPI, 1:5000, Invitrogen, Thermo Fisher Scientific, Waltham, MA, USA) at room temperature for 2 h. After 3 × 5-min PBS washes, sections were mounted onto slides using ProLong™ Gold Antifade Mountant (Invitrogen, Thermo Fisher Scientific, Waltham, MA, USA), air-dried, and cover-slipped.

### 4.4. Imaging and Analysis

The immunostained sections were observed and imaged on a confocal system (Leica SP8 or SPE confocal microscope) (Leica, Wetzlar, Germany). The images for the dentate gyrus marker analysis were taken using a 20 × 1.0 N.A. objective with an additional ×0.75 magnification lens in line. All single-immunolabeled and double-immunolabeled signals/cells in this area were quantitatively analyzed using ImageJ (1.52n 22, NIH, Bethesda, MD, USA). The sections were processed in parallel, and images were acquired using identical settings to allow for comparable measurements. These data were then normalized to the area of the dentate gyrus granule layer defined by DAPI staining. Five sections were picked from each animal, evenly distributed through the anterior to posterior axis of the dentate gyrus.

For analysis of dendritic development, three-dimensional (3D) reconstructions of entire dendritic processes of each GFP + neuron were obtained from Z-series stacks of confocal images using an excitation wavelength of 488 nm at magnification of 20 × 1.0 N.A. objective with an additional ×0.75. All GFP + DG neurons with largely intact, clearly identifiable dendritic trees were analyzed for total dendritic length using Neurolucida (Williston, VT, USA). The measurements did not include corrections for inclinations of dendritic process and therefore represented projected lengths. Sholl analysis for dendritic complexity was carried out by counting the number of dendrites that crossed a series of concentric circles at 10 μm intervals from the cell soma using Neurolucida Explorer (Williston, VT, USA). At least 15 neurons were picked randomly from each animal [32].

For completing 3D reconstruction of spines, consecutive stacks of images were acquired using an excitation wavelength of 488 nm at high magnification (63 × 1.0 N.A. objective with an additional ×10 magnification lens) to capture the full depth of dendritic fragments (30 ± 10 μm long, 60–80 dendritic fragments in each condition analyzed) and spines using a confocal microscope (Leica SP8 confocal microscope, Leica, Wetzlar, Germany). Confocal image stacks were deconvoluted using a blind deconvolution method (Autoquant X; Media Cybernetics, Rockville, MD, USA). The structure of dendritic fragments and spines was traced using 3D Imaris software using a “fire” heatmap and a 2D x–y orthoslice plane to aid visualization (9.2, Bitplane, Belfast, UK). Dendritic fragments were traced using an automatic filament tracer, whereas dendritic spines were traced by means of an autopath method with a semiautomatic filament tracer (diameter; min: 0.1, max: 2.0, contrast: 0.8).

For spine classification, a custom MatLab (MathWorks, Natick, MA, USA) script was used based on an algorithm of stubby: length (spine) < 1.5 and max width (head) < mean_width (neck) × 1.2; mushroom: max width (head) > mean width (neck) × 1.2 and max_width (head) > 0.3. If the spine were not classified as mushroom or stubby, it was defined as long–thin. Axonal bouton volume from axonal fragments was measured using 3D Imaris software and using a magic wand menu (Bitplane, Belfast, UK) after deconvolution. At least 20 spine segments were picked randomly from each animal. The region of the dendrite that was analyzed was a segment 20 to 30 μm from the nuclear rim according to DAPI staining.

### 4.5. Statistical Analysis

Results are expressed as means ± SD. Statistical analysis was conducted using Prism 8.0 Software (GraphPad Inc., San Diego, CA, USA). IHC data were analyzed using one-way ANOVA with multiple comparisons followed by post hoc Tukey test. For Sholl analysis, one-way ANOVA with Dunnett’s multiple-comparison test was used at each point to test for differences between distributions. Y-maze data were analyzed using two-tailed Student’s *t*-test, and fear-conditioning data were analyzed using one-way ANOVA with multiple comparisons followed by Tukey post hoc test. The sample sizes are indicated in the figure legends and were chosen based on experience or published literature. The data that were examined with the parametric test were determined to be normally distributed, and the criteria for statistical significance was set a priori at *p* < 0.05.

## Figures and Tables

**Figure 1 ijms-25-06743-f001:**
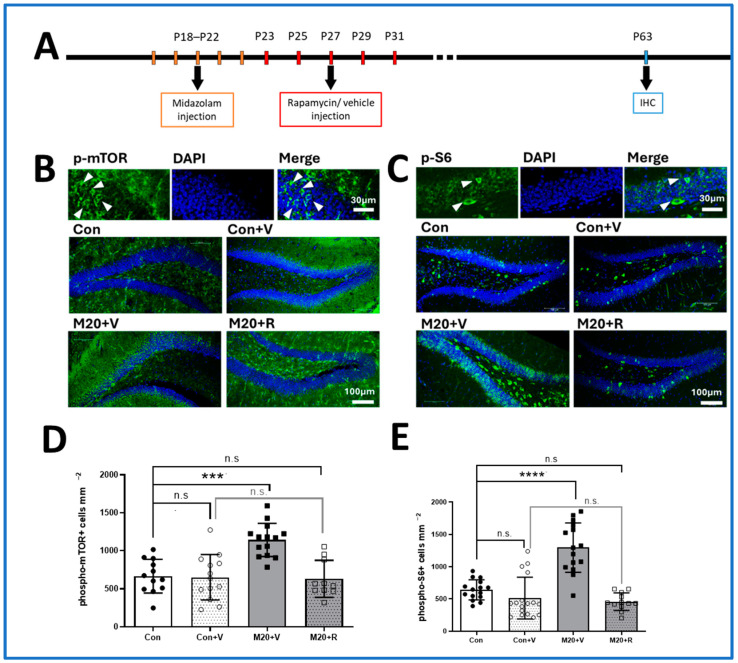
Midazolam exposure leads to aberrant activation of the mTOR signaling pathway in the mouse dentate gyrus. (**A**). Schematic representation of the experimental timeline for immunohistochemistry. (**B**,**C**). Representative confocal images of phospho-S6 and phospho-mTOR expression in exposed dentate gyrus neurons at P63 (scale bar = 100 μm). The upper rows represent the magnified confocal images, and the white arrows point at the qualified positive cells (scale bar = 30 μm). (**D**,**E**). Downstream markers of the mTOR pathway, phospho-mTOR and phospho-S6 expression increased after midazolam exposure, and using the inhibitor of mTOR pathway can prevent this increase. M20 indicates midazolam 20 mg/kg dose administered in deep sedation paradigm using validated rating scale. V indicates rapamycin control vehicle (n = 12–17 per group. *** *p* < 0.001, **** *p* < 0.0001, ANOVA, n.s. indicates no significant difference compared to the control group).

**Figure 2 ijms-25-06743-f002:**
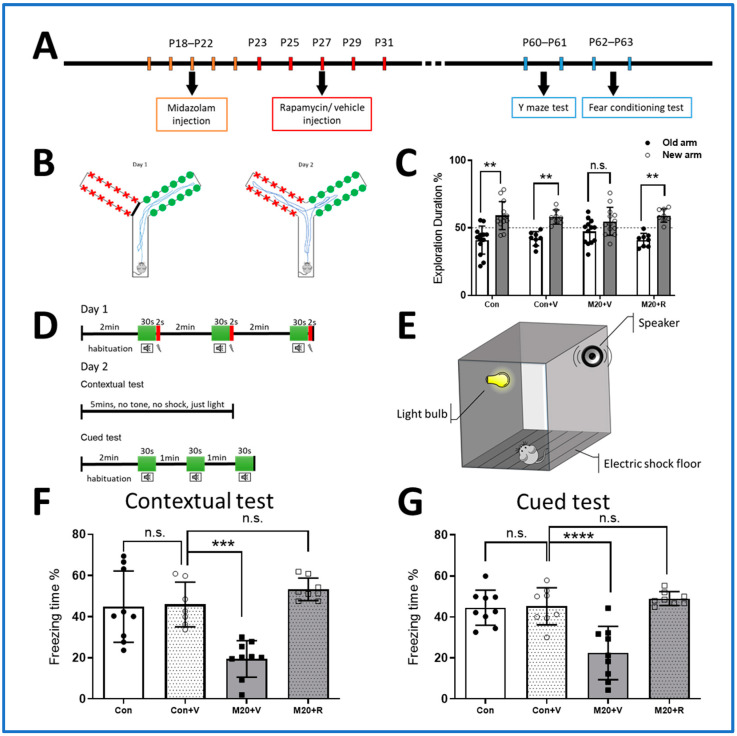
Inhibition of mTOR activities rescues deficits in dentate gyrus-dependent spatial learning and memory after midazolam exposure. (**A**). Schematic representation of the experimental timeline for behavior tests (**B**,**C**). In the Y-maze test (diagram in (**B**)), control animals exhibited normal performance, spending significantly longer times in the novel arm. Midazolam-sedated animals (both low- and high-dose groups) did not differ significantly in time spent in the novel and old arms, suggesting impaired learning and memory. After rapamycin treatment, the results returned to normal (**B**) (n = 8–13 per group, ** *p* < 0.01, *t*-test, n.s. indicates no significant difference compared to the control group). (**D**–**G**). In the fear-conditioning test (diagram in (**D**,**E**)), compared to controls, the midazolam-sedated mice showed significantly reduced freezing time percentages in the contextual test (**F**) and the cued test (**G**). After rapamycin treatment, the midazolam-exposed animals showed no significant different compared to the control group (n = 7–9 per group. *** *p* < 0.001, **** *p* < 0.0001 ANOVA, n.s. indicates no significant difference).

**Figure 3 ijms-25-06743-f003:**
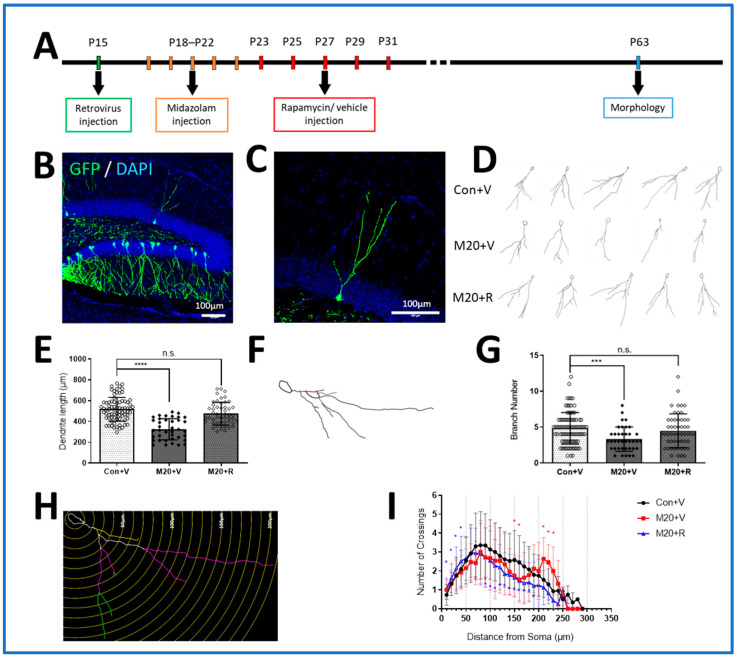
Effects of midazolam exposure on the arborization of newborn neurons in the mouse dentate gyrus. (**A**). Schematic representation of the experimental timeline for morphology examination. (**B**,**C**). Representative confocal images of newborn DG neurons infected with GFP + virus (imaged at P63). (**D**–**G**). Quantification of the total dendritic length and branch numbers of unexposed control, midazolam-exposed, and midazolam-exposed + rapamycin-treated newborn DG neurons at P63, (*** *p* < 0.001, **** *p* < 0.0001, ANOVA, n.s. indicates no significant difference compared to the Con+V group). Numbers on the bar graphs indicate the number of neurons examined from at least 5 animals per group. (**H**,**I**). Sholl analysis of DG neurons after different treatments at P63, (* *p* < 0.05, ANOVA. The red asterisk indicates the comparison between the M20 + V and Con + V group; the blue asterisk indicates the comparison between the M20 + R and Con + V group).

**Figure 4 ijms-25-06743-f004:**
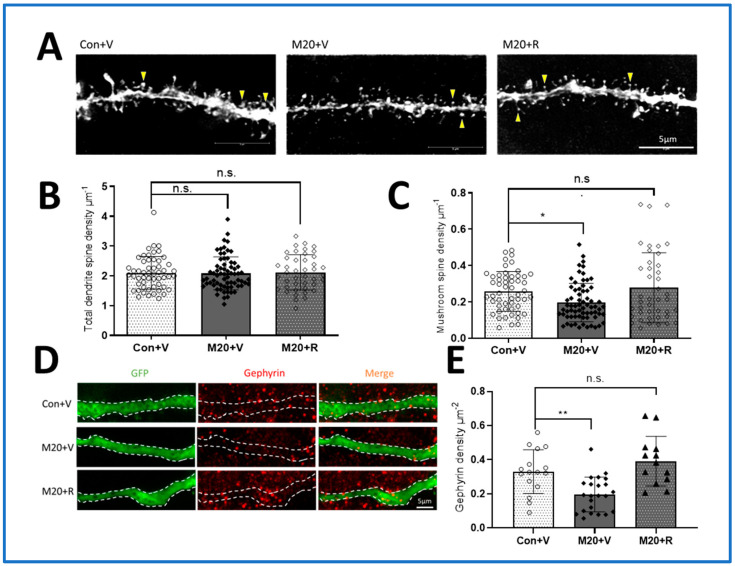
Midazolam sedation causes an mTOR-dependent loss of both mature mushroom morphology excitatory synapses and inhibitory synapses. (**A**). Representative confocal images of newborn neuron dendrites and spines infected with GFP + virus to define dendrites (imaged at P63). Yellow arrows point at the mature mushroom spines. (**B**,**C**). Quantification of the total dendritic spines and mature spines of unexposed control, midazolam-exposed, and midazolam-exposed + rapamycin-treated neurons. Numbers on the bar graph indicate the number of dendritic segments examined from at least 5 mice from each group (* *p* < 0.05, ANOVA, n.s. indicates no significant difference compared to the Con + V group). (**D**). Representative images of gephyrin (green) and GFP (red) immunofluorescence of newborn DG neuron dendrites infected with GFP + virus (imaged at P63). (**E**). Quantification of the puncta number on gephyrin labeling in the unexposed control, midazolam-exposed, and midazolam-exposed+ rapamycin-treated DG neurons. Numbers on the bar graph indicate the number of dendritic segments examined from at least 5 mice from each group (** *p* < 0.01, ANOVA, n.s. indicates no significant difference compared to the Con + V group).

**Figure 5 ijms-25-06743-f005:**
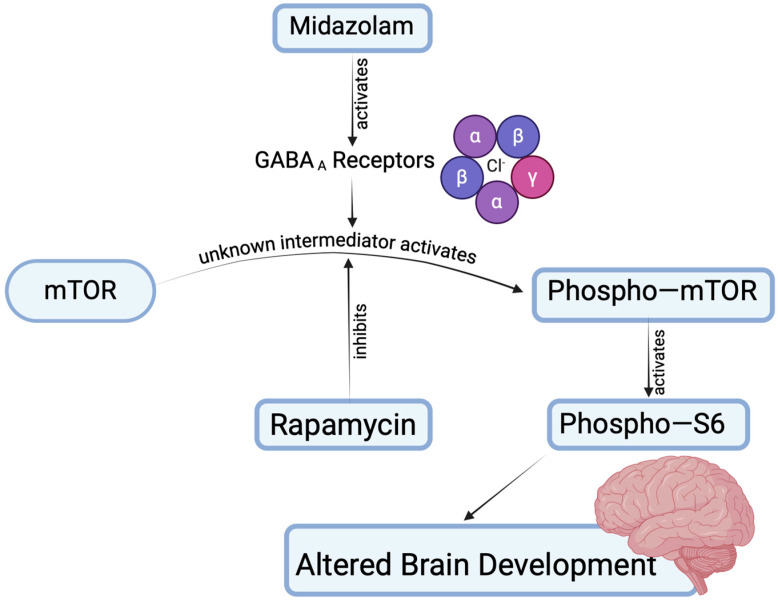
Graphical representation of the study findings and the mTOR pathway. This figure illustrates the key findings of our study in relation to the mTOR pathway. Arrows indicate pathway directions.

## Data Availability

Raw data were generated at Johns Hopkins University. Derived data supporting the findings of this study are available from the corresponding author C.D.M. on request.

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
