# Peer review of "Early Postnatal Exposure to Midazolam Causes Lasting Histological and Neurobehavioral Deficits via Activation of the mTOR Pathway"

_ijms, 2024, doi:10.3390/ijms25126743_

Round 1

Reviewer 1 Report

Comments and Suggestions for Authors

To the editor,

International Journal of Molecular Sciences,

 Thanks for the invitation to review this manuscript IJMS 3018241. In this article, titled as ‘Early postnatal exposure to midazolam in mice causes lasting behavioral deficits and disrupts development of newborn hippocampal neurons via activation of the mTOR pathway.’, Jing Xu and colleagues studied the effect of Midazolam, a commonly used sedative in pediatric intensive care, on neuronal development and subsequent cognitive function via actions on the mechanistic target of rapamycin (mTOR) pathway. Their assays used the mice model system. The mice in the midazolam sedation group showed significant increase in the expression of mTOR activity pathway markers in comparison to controls. Moreover, both neurobehavioral outcomes, including deficits in Y-maze and fear conditioning performance, and neuropathologic effects of midazolam sedation exposure, including disrupted dendritic arborization and synaptogenesis, were ameliorated via treatment with rapamycin, a pharmacologic mTOR pathway inhibitor. My vote is that your journal should not accept this study in its present form. The authors need to do more replicates of the assays and show more drastic changes in midazolam treated mice. My comments and concerns are mentioned below.  

·       Figure 1: This figure contributes significantly to this manuscript. However, the images look very disorganized, and legends are not clearly mentioned. Do the bottom panel of B and C depict zoomed images? Authors need to clarify if the difference between midazolam 10 mg/kg and 20 mg/kg sample is significant and if not, why they are not.

·       Figure 2: The difference between sample con+v and M10+v is minimal in figure 2F and 2G. The authors need more data for the M10+v sample.

·       Figure 3: How the difference between con+v and M20+v is physiologically significant in 3E and 3F? Authors should comment on that. In my opinion, the difference in branch number is minimal to conclude anything. In 3E, the Y axis should be ‘Dendrite length’.

·       Figure 4: I cannot agree with the authors on the inference of the figure. Other than data in figure 4E, the data does not show that exposure to midazolam affects spine density and GABAergic synaptic marker Gephyrin of the newborn neurons. Like I stated above, the difference between con+v and M20+v cannot be physiologically significant.

·       Some acronyms like DG and DCG need to be clarified in the text.

·       Figure legends should be more descriptive and should elaborate the sample names in the figures more clearly.

Author Response

  1. Figure 1: This figure contributes significantly to this manuscript. However, the images look very disorganized, and legends are not clearly mentioned. Do the bottom panel of B and C depict zoomed images? Authors need to clarify if the difference between midazolam 10 mg/kg and 20 mg/kg sample is significant and if not, why they are not.

Thank you so much for your comments.  

There is no significant difference between the lower and higher dose of midazolam in our current results.  We revised the figures according to all the reviewers’ suggestions. For the current figure 1, the top panel of B and C depict zoomed images, we clarified that in the scale bars and legends. We have also revised the legend associated and clearly indicated which images are zoomed. We marked it red in the legends as “The upper rows represent the magnified confocal images, white arrows point at the qualified positive cells (scale bar= 30μm).” 

After consideration of Reviewer 1’s commentary, we have decided to remove the lower dose midazolam results from the main body of the study in response to this critique. As the reviewer has pointed out, differences between the smaller dose and other conditions are not large when they are significant at all. From a practical perspective, we do not think it is feasible to conduct the large number of replicates that might demonstrate a significant difference between low and high dose. Any such difference would likely be small, thus not justifying the large number of animals and the expense required, even if there is a difference.  Thus, we have moved all mention of the lower dose results to the supplemental section so readers can have access to the data but be aware of its limitations, but it is no longer a focal point of the manuscript. 

  1. Figure 2: The difference between sample con+v and M10+v is minimal in figure 2F and 2G. The authors need more data for the M10+v sample.]

As noted above, we have moved all references to the lower dose of midazolam to the supplementary section.  

  1. Figure 3: How the difference between con+v and M20+v is physiologically significant in 3E and 3F? Authors should comment on that. In my opinion, the difference in branch number is minimal to conclude anything. In 3E, the Y axis should be ‘Dendrite length’.

We appreciate the reviewer catching the mistake in the 3E Y axis label, this has been corrected as requested. 

In this measure and also in some others, we realize based on this thoughtful critique that we should have presented the magnitude of the change more clearly so that its significance would be more clearly evident to the reader.  The percentage difference in branching and dendrite length which we measured are 30.95% and 37.18% respectively, and in the revised text we have presented these changes in magnitude as a percentage from baseline to highlight that they are likely to be physiologically meaningful based on the current understanding of the literature on dendrite development. While there is no clear understanding in the literature of the exact threshold for what magnitude of change in measures of dendrite growth correlates to specific functional outcomes, numerous published works have considered changes that are of similar or even lesser magnitude to be physiologically significant. To highlight this point, we have noted this in the result section and added three example references from different areas of study which illustrate changes in dendrite development similar or less than our own that have been seen as physiologically significant.  

  1. Figure 4: I cannot agree with the authors on the inference of the figure. Other than data in figure 4E, the data does not show that exposure to midazolam affects spine density and GABAergic synaptic marker Gephyrin of the newborn neurons. Like I stated above, the difference between con+v and M20+v cannot be physiologically significant.

As in the critique immediately above, we appreciate the reviewer’s highlighting an easily corrected deficiency in our original manuscript. We have substantially clarified the magnitude of change by expressing it in terms of percentage change from baseline. The results now reflects that we measured a 23.1% decrease in mushroom morphology spines.  Given that this spine category is thought to be primarily responsible for the storage of long term memories, the loss of almost a quarter of this computational power is likely to be highly significant as an explanation for the behavioral changes we observed which are dependent on intact function of the hippocampal circuitry.  The revised text also now reflects that we observed a 40.56% loss in inhibitory synapses, as measured by the density of gephyrin-positive puncta, which we also consider to be potentially significant, although the linkage between inhibitory synapse number and learning functions is not as well established.  In parallel with the changes to text reflecting dendrite growth, we have added text to the results section which includes example references of other published works in which changes of similar or even lesser magnitude have been established as physiologically significant.  

  1. Some acronyms like DG and DCG need to be clarified in the text.

We appreciate the reviewer for noting this problem and have corrected this oversight in the revised manuscript. 

  1. Figure legends should be more descriptive and should elaborate the sample names in the figures more clearly.

We have updated figure legends to be more fully descriptive of the associated data visualizations. 

Reviewer 2 Report

Comments and Suggestions for Authors

The study investigates the effects of midazolam, a commonly used sedative in pediatric intensive care, on brain development. Specifically, it examines whether prolonged and repetitive exposure to midazolam interferes with neuronal development and cognitive function through the mechanistic target of the rapamycin (mTOR) pathway. The study concludes that prolonged and repetitive exposure to midazolam sedation during brain development interferes with the formation of neural circuitry by pathologically increasing mTOR pathway signaling. This disruption has lasting negative consequences on both brain structure and function. There are a few matters which need to be addressed before publication. 

Comment for authors

1. The title is too long and wordy. Consider revising it.

2. The background information provided in the introduction section is insufficient for comprehensively understanding the importance and significance of the study subject. I recommend revising the introduction to provide a clearer and more comprehensive overview of the context and relevance of the research topic.

3. What are the mechanistic differences in how midazolam affects dendritic arborization and synaptogenesis compared to other commonly used pediatric sedatives? Explain in the discussion section.

4. It would be beneficial for the authors to include a graphical representation of their findings and the mTOR pathway in the discussion section to enhance understanding.

5. The paper contained typos and grammatical errors. Double-check and correct them in the revised version.

Comments on the Quality of English Language

The paper contained typos and grammatical errors. Double-check and correct them in the revised version.

Author Response

  1. The title is too long and wordy. Consider revising it.

Thank you for your comments. We have revised our title to: 

“Early postnatal exposure to midazolam causes lasting histologic and neurobehavioral deficits via activation of the mTOR pathway”. 

  1. The background information provided in the introduction section is insufficient for comprehensively understanding the importance and significance of the study subject. I recommend revising the introduction to provide a clearer and more comprehensive overview of the context and relevance of the research topic.

We appreciate the importance of this critique and have revised the background section accordingly to attempt to give the reader a fuller understanding of the potential importance of sedation neurotoxicity as it relates to brain development.  This is rendered somewhat challenging as the field of sedation neurotoxicity in development is quite new and has barely been studied (as opposed to anesthetic toxicity, which has been studied for almost two decades).  We have revised the introduction substantially to address this concern and have referenced the three studies that specifically address midazolam sedation developmental neurotoxicity as part of these changes. 

  1. What are the mechanistic differences in how midazolam affects dendritic arborization and synaptogenesis compared to other commonly used pediatric sedatives? Explain in the discussion section.

We agree that this is a critical question to address in terms of the putative clinical relevance of this work.  Unfortunately given the relative newness of the field of developmental sedation neurotoxicity there is no data to answer this question. However, it is very plausible that dexmedetomidine, the clearest alternative to midazolam sedation in the PICU setting, is likely relatively benign in terms of brain health.  To address the reviewer’s comment, we have added to the discussion by citing references showing that dexmedetomidine is the most likely alternative and that based on work done in anesthesia toxicity it has the potential for reduced or no toxicity in relation to brain development.  

  1. It would be beneficial for the authors to include a graphical representation of their findings and the mTOR pathway in the discussion section to enhance understanding.

We appreciate this comment and have added an additional figure (figure 5) which is a graphical representation of our basic findings as requested. 

  1. The paper contained typos and grammatical errors. Double-check and correct them in the revised version.

We have detected and corrected typos and grammatical errors throughout the manuscript as requested. 

Reviewer 3 Report

Comments and Suggestions for Authors

1.        The figures are too small. Please make them larger.

a.        The graphs are big enough, but the fluorescence images and fonts should be larger.

b.        In the figure for maze test, please indicate what are the red and green.

c.        Also, in some point, the figure indication is not correct (ex. line121). Please fix those errors.

2.        Did the authors measure how much amount of midazolam in the mice? How long the midazolam stayed in animals?

3.        The authors performed the behavior test and observe the morphologies of dendate gyrus neuron with P63, however, is there any effects of midazolam on young mice?

4.        In lines 184-188, the authors reported as “there’s no significant difference in the total density of the spines in the midazolam-exposed group(2.08± 0.56/μm) and the control group (2.10± 0.55/μm, p=0.9876), while there was a significant decrease in the density of the mature mushroom spines (M20+V: 0.20± 0.11/μm vs CON+V: 0.10± 0.11/μm, p=0.0241) (Fig 4B, C)”. However, in Figure 4B and C, it is difficult to mention that there is a significant decrease in the density of the mature mushroom spine. Please discuss this in more detail.

Author Response

Comments and Suggestions for Authors 

  1. The figures are too small. Please make them larger.
  2. The graphs are big enough, but the fluorescence images and fonts should be larger.

We appreciate the importance of this concern and have increased the size of the fluorescence images and the figure fonts as requested.  

  1. In the figure for maze test, please indicate what are the red and green.

We agree that this was inadequately clear.  In this paradigm, red and green indicate distinct visual cues.  To address this concern we have added text in the methods section specifically stating that each arm choice contained an overt visual cue, along with references to previous usage of this testing paradigm:  

  1. Also, in some point, the figure indication is not correct (ex. line121). Please fix those errors.

We appreciate this critique and have double checked and corrected figure references throughout the text. 

  1. Did the authors measure how much amount of midazolam in the mice? How long the midazolam stayed in animals?

We did not measure midazolam levels in blood or CSF, but rather relied on weight-based dosing and a validated sedation scale, which are the two determinants for clinical dosing of sedative medications.  The sedation scale allowed us to compensate for the modest variation in duration of action between mice, which presumably is due to subtle differences in uptake/metabolism/elimination. To address the reviewer’s concern, we have incorporated references and additional text on the pharmacology of midazolam in rodents in the methods section to bring greater clarity to readers. 

  1. The authors performed the behavior test and observe the morphologies of dendate gyrus neuron with P63, however, is there any effects of midazolam on young mice?

We focused our work on the toxic effects of midazolam exposure on the long-term outcomes because many changes in younger animals can be overcome by compensation during development.  There are already some studies of the acute neurological effects of midazolam on young animals and we have incorporated this into the text of the introduction with appropriate references. 

  1. In lines 184-188, the authors reported as “there’s no significant difference in the total density of the spines in the midazolam-exposed group(2.08± 0.56/μm) and the control group (2.10± 0.55/μm, p=0.9876), while there was a significant decrease in the density of the mature mushroom spines (M20+V: 0.20± 0.11/μm vs CON+V: 0.10± 0.11/μm, p=0.0241) (Fig 4B, C)”. However, in Figure 4B and C, it is difficult to mention that there is a significant decrease in the density of the mature mushroom spine. Please discuss this in more detail.

We appreciate the reviewer’s concern. This finding is consistent with previous published work on anesthesia toxicity, in which total spine density was not changed, but subgroup analysis of mushroom morphology spines only showed a significant decrease with anesthesia exposure.  We postulate that a similar process of toxicity is responsible for the loss of the mushroom spines with sedation exposure and given that the mushroom spines are the group that is most critical for the plasticity underlying learning and memory we hypothesize that this change is critical for the neurobehavioral outcomes that we have observed.  To clarify this further for the reader, we have noted loss of mushroom spines as a percentage in the text to better show the magnitude of the change. Additionally, we are including references that discuss the role of the mushroom morphology spine population and also showing that similar changes in mushroom spine populations have been deemed responsible for changes in learning and memory function in other model systems.  Also to further address this concern we have added text in the discussion in which we postulate that the change in mushroom spines without a difference in total spines is likely balanced off by an increase in the other morphology types, which, if true, would represent a shift to more immature spine types. 

Round 2

Reviewer 1 Report

Comments and Suggestions for Authors

Dear editor, thank you for inviting me to review this revised version of this manuscript. I have gone through their result and discussion sections in the main text and figure legends. My opinion is to accept this paper in its present form if other reviewers have no other comments, suggestions or concerns.

Thanks once again.

Best wishes,

Reviewer 2 Report

Comments and Suggestions for Authors

I recommend accepting the paper for publication in its present form.

View Synonyms and Definitions